

# Applications of the indole-alkaloid gramine modulate the assembly of individual members of the barley rhizosphere microbiota

Mauro Maver[1,2,3], Carmen Escudero-Martinez[1], James Abbott[4], Jenny Morris[5], Pete E. Hedley[5], Tanja Mimmo[2,3] and Davide Bulgarelli[1]

[1] Plant Sciences, School of Life Sciences, University of Dundee, Dundee, United Kingdom
[2] Faculty of Science and Technology, Free University of Bozen-Bolzano, Bolzano, Italy
[3] Competence Centre for Plant Health, Free University of Bozen-Bolzano, Bolzano, Italy
[4] Data Analysis Group, School of Life Sciences, University of Dundee, Dundee, United Kingdom
[5] Cell and Molecular Sciences, The James Hutton Institute, Dundee, United Kingdom

## ABSTRACT

Microbial communities proliferating at the root-soil interface, collectively referred to as the rhizosphere microbiota, represent an untapped beneficial resource for plant growth, development and health. Integral to a rational manipulation of the microbiota for sustainable agriculture is the identification of the molecular determinants of these communities. In plants, biosynthesis of allelochemicals is centre stage in defining inter-organismal relationships in the environment. Intriguingly, this process has been moulded by domestication and breeding selection. The indole-alkaloid gramine, whose occurrence in barley (*Hordeum vulgare* L.) is widespread among wild genotypes but has been counter selected in several modern varieties, is a paradigmatic example of this phenomenon. This prompted us to investigate how exogenous applications of gramine impacted on the rhizosphere microbiota of two, gramine-free, elite barley varieties grown in a reference agricultural soil. High throughput 16S rRNA gene amplicon sequencing revealed that applications of gramine interfere with the proliferation of a subset of soil microbes with a relatively broad phylogenetic assignment. Strikingly, growth of these bacteria appeared to be rescued by barley plants in a genotype- and dosage-independent manner. In parallel, we discovered that host recruitment cues can interfere with the impact of gramine application in a host genotype-dependent manner. Interestingly, this latter effect displayed a bias for members of the phyla Proteobacteria. These initial observations indicate that gramine can act as a determinant of the prokaryotic communities inhabiting the root-soil interface.

# INTRODUCTION

The interface between roots and soil hosts distinct microbial communities, collectively referred to as rhizosphere microbiota (*Turner, James & Poole, 2013*). These plant-microbial assemblages define a continuum of parasitic, commensal, and mutualistic

Corresponding authors
Tanja Mimmo, tmimmo@unibz.it
Davide Bulgarelli,
d.bulgarelli@dundee.ac.uk

interactions (*Schlaeppi & Bulgarelli, 2015*). For example, so-called plant growth-promoting rhizobacteria can enhance plant mineral uptake and protect their hosts from pathogens (*Lugtenberg & Kamilova, 2009*). Studies conducted with multiple plant species support the notion that members of the rhizosphere microbiota are not passively assembled from the surrounding soil biota, rather a multi-step selection process differentiate plant-associated communities from those identified in unplanted soils (*Bulgarelli et al., 2013*; *Edwards et al., 2015*). This selection process operates at multiple taxonomic and functional ranks, with the enrichment of members of the phyla Actinobacteria, Bacteroidetes, Proteobacteria and Firmicutes representing the distinct signature of plant-associated communities (*Bulgarelli et al., 2013*; *Alegria Terrazas et al., 2016*). The host genome represents one of the determinants in the selection process defining the microbiota thriving at the root-soil interface (*Schlaeppi & Bulgarelli, 2015*; *Hacquard et al., 2015*; *Alegria Terrazas et al., 2016*). This selection is exerted through a number of plant traits, including root system architecture (*Robertson-Albertyn et al., 2017*; *Pérez-Jaramillo et al., 2017*) and the plant immune system (*Lebeis et al., 2015*). Another key element of the host-mediated molecular mechanisms shaping the rhizosphere microbiota is the root exudation of metabolites capable of modulating the interactions among plants, microbes and the surrounding environment (*Dakora & Phillips, 2002*; *Jones, Nguyen & Finlay, 2009*; *Pascale et al., 2020*). Consistently, an increasing number of plant primary (*Canarini et al., 2019*), as well as secondary (*Rolfe, Griffiths & Ton, 2019*) metabolites have recently been implicated in shaping the plant microbiota.

Modern crops are the result of an on-going selection process, initiated with domestication and continued with breeding selection, which progressively differentiated cultivated varieties from their wild ancestors (*Purugganan & Fuller, 2009*). Interestingly, these selection processes impacted on both plant's ability to assemble a rhizosphere microbiota (*Pérez-Jaramillo, Mendes & Raaijmakers, 2016*) and its capacity of secreting metabolites at the root-soil interface (*Preece & Peñuelas, 2020*).

As wild ancestors of modern cultivated varieties may hold the capacity to adapt to marginal soil conditions, there is a growing interest in discerning the molecular mechanisms underpinning microbiota recruitment in crop wild relatives and their contribution to plant's adaptation to the environment (*Escudero-Martinez & Bulgarelli, 2019*). This is particularly attractive for crops like barley (*Hordeum vulgare* L.), the fourth most cultivated cereal worldwide, for which modern and wild genotypes are readily available for experimentation (*Bulgarelli et al., 2015*; *Alegria Terrazas et al., 2020*).

The genus *Hordeum* has evolved two main indole alkaloids with allelopathic and defensive functions, the benzoxazinoid DIBOA and gramine, whose biosynthesis appear mutually exclusive within barley lineages (*Grün, Frey & Gierl, 2005*). In particular, gramine is the main allelochemical of the lineage *H. vulgare* which has historically been implicated in defensive responses against insects (*Corcuera, 1993*; *Cai et al., 2009*; *Sun et al., 2013*), as well as foliar pathogens (*Sepulveda & Corcuera, 1990*; *Matsuo et al., 2001*) although the genetic basis of this trait appears complex (*Åhman, Tuvesson & Johansson, 2000*; *Macaulay, Ramsay & Åhman, 2020*). Intriguingly, crop selection left a footprint on the biosynthesis of this secondary metabolite: modern cultivated, so called 'elite', varieties (*H. vulgare*

subp. *vulgare*) often fail to accumulate gramine to levels identified in their wild relatives (*H. vulgare* subp. *spontaneum*) (*Matsuo et al., 2001*; *Maver et al., 2020*). Of note, this apparent counter-selection for gramine within domesticated material has been exerted on at least two distinct biosynthetic genes in barley (*Larsson et al., 2006*).

To gain novel insights into the ecological significance of gramine for plant-microbiota interactions, we hypothesized that the release of this secondary metabolite acts as a recruitment cue for the barley microbiota. Despite F1 hybrids between wild and domesticated genotypes producing gramine do exist (*Moharramipour et al., 1999*), no barley isogenic lines for the biosynthesis of this secondary metabolites are currently accessible to experimentation. This makes it difficult to discriminate between gramine and other genetic factors putatively impacting on microbiota composition. We therefore decided to test our hypothesis, by exposing two 'elite', gramine-free, barley genotypes, the cultivars Morex and Barke (*Larsson et al., 2006*), to exogenous applications of gramine and we assessed the impact of these treatments on the taxonomic composition of the prokaryotic microbiota thriving at the root-soil interface using a cultivation independent approach.

## MATERIALS & METHODS

### Soil substrate

The experiments were carried out in "Quarryfield" soil, an unfertilized Scottish agricultural soil collected at the site of The James Hutton Institute, Invergowrie, Scotland (UK) (56°27′5″N 3°4′29″W). This soil was used previously to grow barley and left unplanted at least 4 years before being used for the experiments. Physical and chemical characterization: Silt: 39.35%; Clay: 11.08%; Sand: 49.58%; pH in water: 5.8; C.E.C.: 15.05 [meq/100g]; Org. Matter DUMAS: 5.95%.

### Gramine adsorption

Five grams of Quarryfield soil were mixed with 10 mL of 10 mmol $L^{-1}$ $CaCl_2$ solution containing gramine at the following concentrations: 0, 1.5, 5, 10, 15, 20, 30, 60, 100, 150, 200, 250, 300, 400, 600, 800, 1,000 mg $L^{-1}$. Five replicates were prepared for each concentration. Soil suspensions were shaken for 24 h at room temperature. Then, suspensions were centrifuged for 5 min at 5,000 g, the supernatant was collected, filtered (0.45 μm, Phenomenex) and analysed by liquid chromatography (HPLC) using the method reported in (*Maver et al., 2020*). Pure reagent grade gramine was used to prepare two stock solutions in water of 0.5 mol $L^{-1}$ and 10 mmol $L^{-1}$. Adsorption solutions have been prepared by dilution from the stock solution with milliQ water. The adsorption rate of gramine in soil was obtained by difference between the initial and final concentration measured in the supernatant by HPLC.

Adsorption isotherms of gramine were fitted applying several nonlinear models: the two-parameter Langmuir and Freundlich isotherms and the three-parameter Sigmoidal Langmuir, Redlich-Peterson and Sips isotherms (*Limousin et al., 2007*; *Foo & Hameed, 2010*).

## Plant material

Seeds of elite barley genotypes, two-row malting Barke and six-row malting Morex were selected for this experiment. Both are well represented in barley studies: Barke as a parental donor in the development of a nested associated mapping (NAM) population (*Maurer et al., 2015*), and Morex for being the first sequenced barley genotype (*The International Barley Genome Sequencing Consortium, 2012*).

## Growth conditions

Barley seeds were cleaned using deionized water and gently shaken for 1 min. After that water was discarded and the whole process repeated 3 times. For seed germination, seeds were placed on petri dishes containing a semi-solid 0.5% agar solution. After a week, seedlings displaying a comparable development were individually transferred into 12-cm pots containing approximately 500 g of Quarryfield agricultural soil (*Robertson-Albertyn et al., 2017*; *Alegria Terrazas et al., 2020*) previously sieved to remove stones and large debris. Plants were grown in a randomized design in a glasshouse at 18/14 °C (day/night) temperature regime with 16 h daylight that was supplemented with artificial lighting to maintain a minimum light intensity of 200 $\mu$mol quanta m$^{-2}$ s$^{-1}$. The stock solution of gramine was prepared by adding pure reagent grade gramine (Sigma-Aldrich, >99%), in water, sonicated for 20 min and then stored at 4 °C. After 4 days of growing in soil, two different final gramine concentrations, 24 $\mu$mol L$^{-1}$ and 46 $\mu$mol L$^{-1}$, were added directly on the soil of selected pots. Mock controls (*i.e.*, gramine 0 $\mu$mol L$^{-1}$) were supplemented with sterilised water. Additional watering was performed every 2 days with the application of 50 mL of deionized water to each pot. For each gramine treatment (*i.e.*, 0, 24 and 46 $\mu$mol L$^{-1}$) we used five replicates (*i.e.*, five individual pots) per barley genotype and unplanted pots containing the same soil substrates used as 'bulk' soil controls (*i.e.*, 45 pots in total). Individual replicated pots were maintained in the glasshouse for 4 weeks post-transplantation, when the tested genotypes reached early stem elongation, corresponding to Zadoks stages 30–35 (*Tottman, Makepeace & Broad, 1979*).

## Rhizosphere fractionation and sampling of soil-grown barley plants and bulk soil

The preparation of material for amplicon sequencing was performed following established protocols (*Robertson-Albertyn et al., 2017*; *Alegria Terrazas et al., 2020*). Briefly, four-week-old barley plants were carefully removed from the soil, and the shoot and root separated. The shoot was dried at 70 °C for 48 h and the dry weight collected. The roots were gently shaken to remove loosely bound soil particles, and the resulting root system and tightly adhered soil, operationally defined as rhizosphere, was further sectioned to retain the uppermost six cm of the seminal root system of each sample. This root material was transferred in a sterile 50 mL falcon tube containing 15 mL of phosphate buffered saline solution (PBS). Samples were then vortexed for 30 s, the soil sedimented for 2–3 mins, and the roots transferred in a new 50 mL falcon tube with 15 mL PBS, in which the samples were vortexed again for 30 s to separate the remaining rhizosphere soil from roots. The roots were discarded, the two falcon tubes were combined in one single falcon tube, now
containing the rhizosphere soil fraction, and then centrifuged at 1,500 g for 20 min. After centrifugation, the supernatant was discarded, and the pellet immediately stored at −80 °C. In the unplanted soil controls (*i.e.*, the bulk soil pots), a portion of soil corresponding to the area explored by roots was collected with a spatula and processed as described for planted soils. Until DNA extraction, all the samples were stored at −80 °C.

## Metagenomic DNA extraction from rhizosphere and bulk soil specimens

Total DNA was extracted from the rhizosphere and unplanted soil samples using FastDNA[TM] SPIN kit for soil (MP Biomedicals, Solon, USA) following the instructions by the manufacturer. To assess the concentration and the quality (260/280 nm and 260/230 nm ratios) of the extracted DNA, a Nanodrop 2000 spectrophotometer (Thermo Fisher Scientific, Waltham, USA) was used and samples were stored at −20 °C until further analysis. Aliquots at a final DNA concentration of 10 ng mL$^{-1}$ were prepared for each sample using sterilized deionized water, and stored at −20 °C.

## 16S rRNA gene amplicon library construction

The amplicon library was generated via a selective PCR amplification of the hypervariable V4 region of the 16S rRNA gene using the PCR primers 515F (5′-GTGCCAGCMGCCG CGGTAA-3′) and 806R (5′-GGACTACHVGGGTWTCTAAT-3′) as previously described (*Robertson-Albertyn et al., 2017*; *Alegria Terrazas et al., 2020*). Briefly, PCR primer sequences were fused with Illumina flow cell adapter sequences at their 5′ termini and the 806R primers contained 12-mer unique 'barcode' sequences to enable the multiplexed sequencing of several samples in a single pool (*Caporaso et al., 2012*).

For each individual bulk and rhizosphere preparations, 50 ng of DNA was subjected to PCR amplification using the Kapa HiFi HotStart PCR kit (Kapa Biosystems, Wilmington, USA). The individual PCR reactions were performed in 20 μL final volume and contained:

- 4 μL of 5X Kapa HiFi Buffer
- 10 μg Bovine Serum Albumin (BSA) (Roche, Mannheim, Germany)
- 0.6 μL of a 10 mM Kapa dNTPs solution
- 0.6 μL of 10 μM solutions of the 515F and 806R PCR primers
- 0.25 μL of Kapa HiFi polymerase

Reactions were performed in a G-Storm GS1 thermal cycler (Gene Technologies, Somerton, UK) using the following programme: 94 °C (3 min), followed by 35 cycles of 98 °C (30 s), 50 °C (30 s) 72 °C (1 min) and a final step of 72 °C (10 min). For each 515F-806R primer combination, a no template control (NTC) was subjected to the same process. To minimize potential biases originating during PCR amplifications, individual reactions were performed in triplicate and 2 independent sets of triplicate reactions per barcode were performed.

To check the amplification and/or any possible contamination, prior to purification, 6 μL aliquots of individual replicates and the corresponding NTCs were inspected on 1.5% agarose gel. Only samples that display the expected amplicon size and no detectable contamination in NTCs on gel were retained for library preparation.

Individual PCR amplicons replicates were then pooled in a single plate, moving each sample to a specific position according to their barcode. They were purified using Agencourt AMPure XP Kit (Beckman Coulter, Brea, USA) with 0.7 µL AmPure XP beads per 1 µL of sample. Following purification, 6 µL of each sample was quantified using PicoGreen (Thermo Fisher Scientific, Watham, USA). Once quantified, individual barcode samples were pooled to a new tube in an equimolar ratio to generate amplicons libraries.

### Illumina 16S rRNA gene amplicon sequencing

Amplicon libraries were supplemented with 15% of a 4 pM phiX solution and run at 10 pM final concentration on an Illumina MiSeq system with paired end $2 \times 150$ bp reads (*Caporaso et al., 2012*) as recommended, to generate FASTQ sequence files for processing and analysis.

### Amplicon sequencing reads processing

Sequencing reads were processed using a customized bioinformatics pipeline as described before (*Terrazas et al., 2019*). Briefly, sequencing reads were subjected to quality assessment using FastQC (*Andrews, 2010*). Amplicon Sequencing Variants (ASVs) were then generated using DADA2 version 1.10 (*Callahan et al., 2016*) and R 3.5.2 (*Team R Development Core, 2018*) following the basic methodology outlined in the 'DADA2 Pipeline Tutorial (1.10)'. Read filtering was carried out using the DADA2 filterAndTrim method, trimming 10bp of sequence from the 5′ of each reads using a truncQ parameter of 2 and maxEE of 2. The remainder of the reads were of high quality so no 3′ trimming was deemed necessary. The dada method was run to determine the error model with a MAX_CONSIST parameter of 20, following which the error model converged after 9 and 12 rounds for the forward and reverse reads respectively. The DADA2 method was then run with the resulting error model to denoise the reads using sample pooling, followed by read merging, using the default minOverlap parameter of 12 bases, then chimera removal using the consensus method. Taxonomy assignment was carried out using the RDP Naive Bayesian Classifier through the 'assign Taxonomy' method, with the SILVA database (*Quast et al., 2012*) version 138, using a minimum bootstrap confidence of 50. The DADA2 outputs were converted to a Phyloseq object (version 1.26.1) (*McMurdie & Holmes, 2013*). ASVs assigned to chloroplast and mitochondria, putatively representing host contamination, as well as ASVs previously identified as putative lab contaminants (*Pietrangelo et al., 2018*) were removed in silico from the original Phyloseq object. Likewise, ASVs lacking taxonomic classification at Phylum level (*i.e.*, classified as "NAs") were removed from the dataset.

As a secondary quality filtering approach (*Bokulich et al., 2013*), we applied an abundance filtering for any given ASVs to be retained in the final dataset of 20 reads in at least 11% of the samples, representing the number of replicates for a given sample type/condition. In this way, an ASV with overall low abundance due to its association with only one given condition would have been retained and analysed in the dataset. Upon completion of this additional filtering step, we retained 6,615,714 reads (min = 25,615; max = 305,408, mean = 147,015.9) representing over 93.5% of the input quality filtered, non-contaminant sequences. Upon completion of this additional filtering step, individual

ASVs were agglomerated at Genus level in Phyloseq. Finally, to control for sample-to-sample differences exceeding a factor of ~10X in sequencing depth (*Weiss et al., 2017*), we downsized the Phyloseq object at 25,000 reads per sample.

## Statistical analysis

Data analysis was performed in R software v 3.5.2. The following R packages were used: Phyloseq v.1.26.1 (*McMurdie & Holmes, 2013*) for Alpha and Beta-diversity indexes; DESeq2 v1.22.2 (*Love, Huber & Anders, 2014*) for the differential analysis of microbial enrichment; ggplot2 v.3.3.2 (*Wickham, 2016*) for data visualization; Vegan v.2.5-6 (*Oksanen et al., 2019*) for statistical analysis of beta-diversity; PMCMR v.4.3 (*Pohlert, 2018*) for non-parametric analysis of variance.

The analysis of the microbiota data was performed on the filtered "Phyloseq object" described above and linking into the analysis the mapping file (metadata information). For the Alpha-diversity analysis, Chao1, Observed ASVs and Shannon indices were calculated using the function to estimate richness included in the Phyloseq package.

For the Beta-diversity analysis, the rarefied ASV table was used as input to compute a Bray–Curtis, dissimilarity matrices. This dissimilarity matrix was visualized using Principal Coordinates Analysis (PCoA) and Canonical Analysis of Principal coordinates (CAP) (*Anderson & Willis, 2003*). Beta-diversity dissimilarity matrices were tested by Permutational Multivariate Analysis of Variance (Permanova) using Adonis function in Vegan package over 5,000 permutations, to calculate the statistical significance.

## Microbial enrichment analysis

The analysis of the microbial enrichment was performed using the DESeq2 package (*Love, Huber & Anders, 2014*), in order to identify the number of genera significantly enriched in pair-wise comparisons with an adjusted $p$ value (False Discovery Rate, FDR $p < 0.05$). The microbial enrichment analysis was carried out between bulk soil and planted soil to evaluate the impact of gramine on the 'rhizosphere effect'. The number of genera enriched in the rhizosphere samples subjected to the three different concentrations of gramine, in both Morex and Barke, was plotted using the package UpSetR (*Conway, Lex & Gehlenborg, 2017*). In parallel, we performed a series of pair-wise comparisons between Morex and Barke using the genera enriched in these latter specimens from unplanted soil controls. Differentially enriched genera were visualized using ternary plots as previously described (*Bulgarelli et al., 2012*).

## RESULTS

### Gramine application impacts on the alpha- and beta-diversity properties of the microbiota thriving at the barley root-soil interface

Gramine availability and mobility in Quarryfield soil are two prerequisites to fulfil its allelopathic ability towards target plants and inter-organismal relationships. Thus, we assessed its adsorption applying several two- and three-parameter non-linearized isotherms models, with the best fit based on the residual sum of squares (RSSs; Fig. S1). This choice was dictated by the fact that non-linear forms permit greater accuracy in predicting

parameters compared to linear forms (*Foo & Hameed, 2010*). The model with the best fit, the Sigmoidal Langmuir isotherm model, resulted in a sigmoidal curve, describing a cooperative adsorption phenomenon, which is common for non-polar compounds (*Sparks, 2003*; *Limousin et al., 2007*). A $K_{foc}$, defined as organic-carbon normalized Freundlich distribution coefficient, estimated at 1390 (*OECD, 2001*) classifies gramine as slightly mobile in this particular soil according to FAO Mobility Classification (*FAO, 2000*), yet remaining available in the soil solution. In addition, no significant differences were recorded by comparing plant dry weight at sampling time (Fig. S2, ANOVA followed by Tukey HSD test, Barke $p$ value = 0.432 and Morex $p$ value = 0.9). These data indicate that Quarryfield soil is a suitable substrate to investigate the impact of the exogenous application of gramine and the root-soil interface and that this latter is not associated to obvious pleiotropic effect on barley growth.

Next, we generated a 16S rRNA gene amplicon library from 45 bulk soil and rhizosphere samples exposed to different concentration of gramine. Upon processing of the sequencing reads in silico (Material and Methods) with a protocol comparable to previous studies conducted in the same soil type (*Robertson-Albertyn et al., 2017*; *Alegria Terrazas et al., 2020*), we failed to identify a significant effect of the treatment on the alpha-diversity parameters of the tested communities (Fig. 1, Kruskal–Wallis test followed by Dunn's post-hoc test, Observed genera $p$ value = 0.5267, Chao1 $p$ value = 0.6657 and Shannon $p$ value = 0.6002, upon Bonferroni correction). Conversely, we observed a clear impact of the external application of gramine on the bacterial communities thriving at the root soil interface regardless of the applied concentration: both a Canonical Analysis of Principal Coordinates (CAP, Fig. 2) and a Principal Coordinates Analysis (PCoA, Fig. S3) built on a Bray–Curtis distance matrix revealed a partition of the microbiota associated to the applied treatment. Of note, the effect of gramine appeared more pronounced on bulk soil samples than rhizosphere specimens. Congruently, a permutational analysis of variance computed on both matrices indicated a significant effect of the individual microhabitat, bulk soil, Morex and Barke rhizosphere, respectively (R2 ~38.6%, Adonis test $p$ value = 0.00020; 5,000 permutations; Table 1) and the interaction term between gramine application and microhabitat (R2 ~6.6; Adonis test $p$ value = 0.01540; 5,000 permutations; Table 1). Conversely, the significance of the impact of gramine *per se* failed marginally to pass the threshold imposed (R2 ~4.9%; Adonis test $p$ value = 0.05559; 5,000 permutations; Table 1). Interestingly, we obtained strikingly similar results when this calculation was performed using individual ASVs (Figs. S4 and S5), suggesting that in the impact of gramine applications on the bacteria proliferating at the root-soil interface is coded by taxonomically conserved portions of their genomes.

## The abundance of individual members of the barley microbiota is affected by gramine application

The observation that rhizosphere profiles tend to converge on the computed ordinations (see Figs. 2 and S3), suggested that the two barley genotypes evolved the capacity of reverting, at least in part, the selective pressure of gramine on soil bacteria. To quantify this phenomenon, we implemented pair-wise comparisons between individual genera

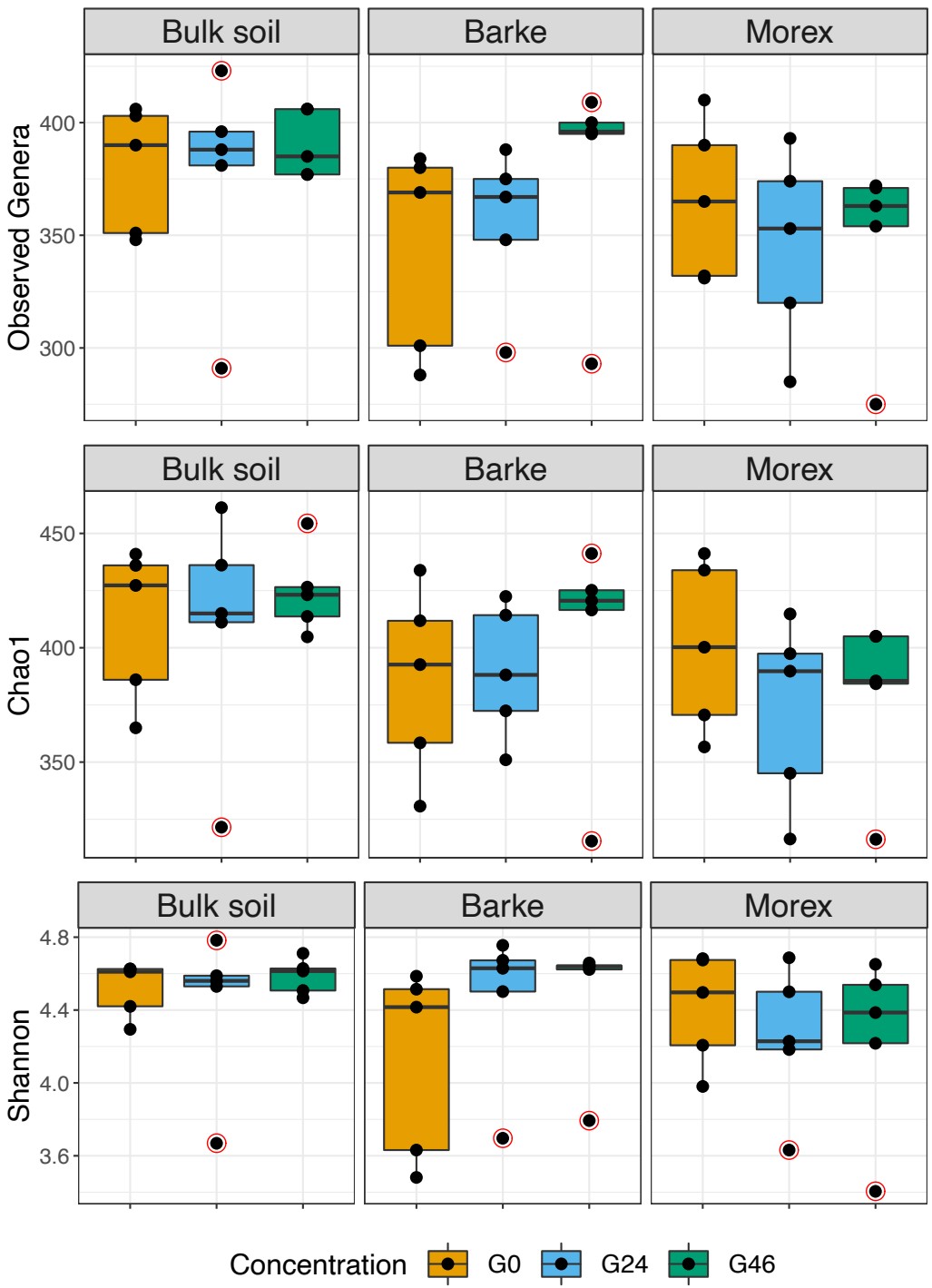

**Figure 1** **Boxplot of alpha diversity indexes.** Alpha diversity Observed genera, Chao1 and Shannon in-dexes, of Bulk soil, Barke and Morex, at three different gramine concentrations. Individual dots depict in-dividual biological replicates; no significant differences observed for gramine treatment upon Kruskal–Wallis test followed by Dunn's post-hoc test, individual $p$ values > 0.05, Bonferroni corrected.

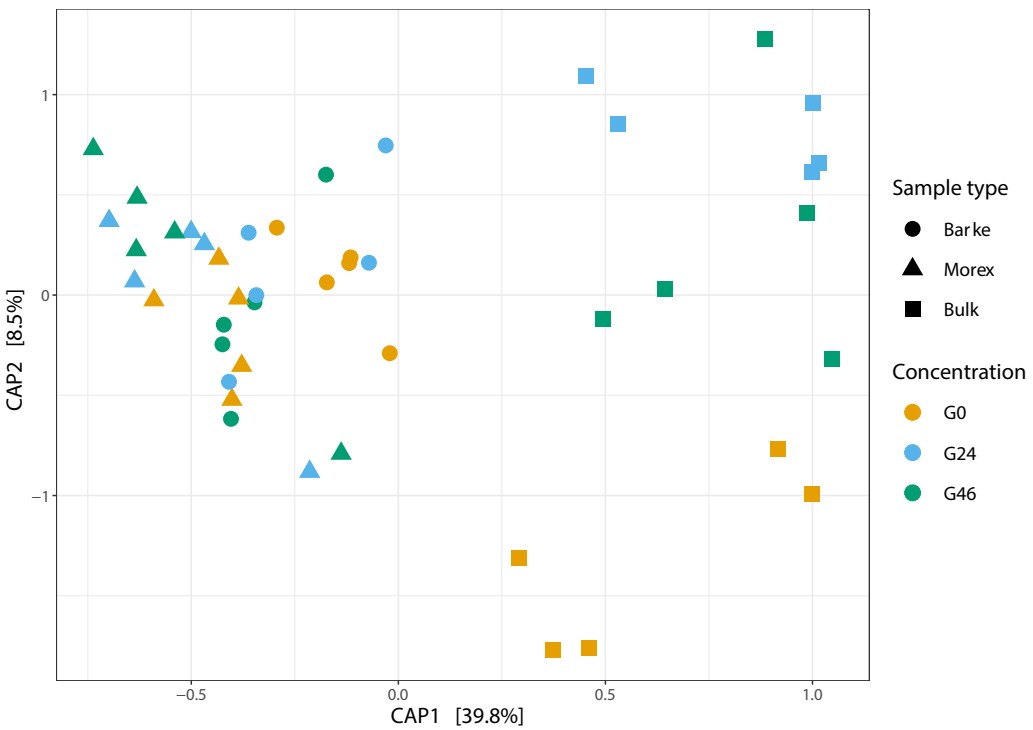

**Figure 2 Canonical analysis of Principal Coordinates (CAP) constructed on a Bray-Curtis dissimilarity matrix of Bulk soil, Barke and Morex, at three different gramine concentrations.** Individual shapes depict individual samples, color-coded according the gramine treatment imposed on them. The ordination was constrained for genotype and gramine concentrations.

**Table 1 Permutational analysis of rhizosphere microbiota variance computed on Bray–Curtis matrix, variance explained by the indicated variables and corresponding statistical significance.**

| Factor Bray–Curtis | R2 | Pr (>F) |
|---|---|---|
| Microhabitat | 0.38595 | 0.00020 |
| Treatment | 0.04911 | 0.05559 |
| Microhabitat:Treatment | 0.06623 | 0.01540 |

retrieved from unplanted soil and rhizosphere communities at the three levels of gramine tested. Congruently with the initial observation, we determined that the majority of genera enriched in the rhizosphere of either genotype are comparable across gramine treatments (Fig. 3, Wald test, individual $p$ values < 0.05, FDR corrected). Conversely, 18 genera, belonging to 16 distinct higher taxonomic ranks, whose cumulative relative abundance represented ~1.66% and ~1.12% of the Morex and Barke rhizosphere communities respectively, were identified as gramine-responsive in a genotype-independent manner (Fig. 4, Wald test, individual $p$ values < 0.05, FDR corrected). Interestingly, we noticed that the host genotype drives the enrichment of genera whose abundance in gramine-treated bulk soil samples was almost obliterated. When looking at the taxonomic affiliation of these
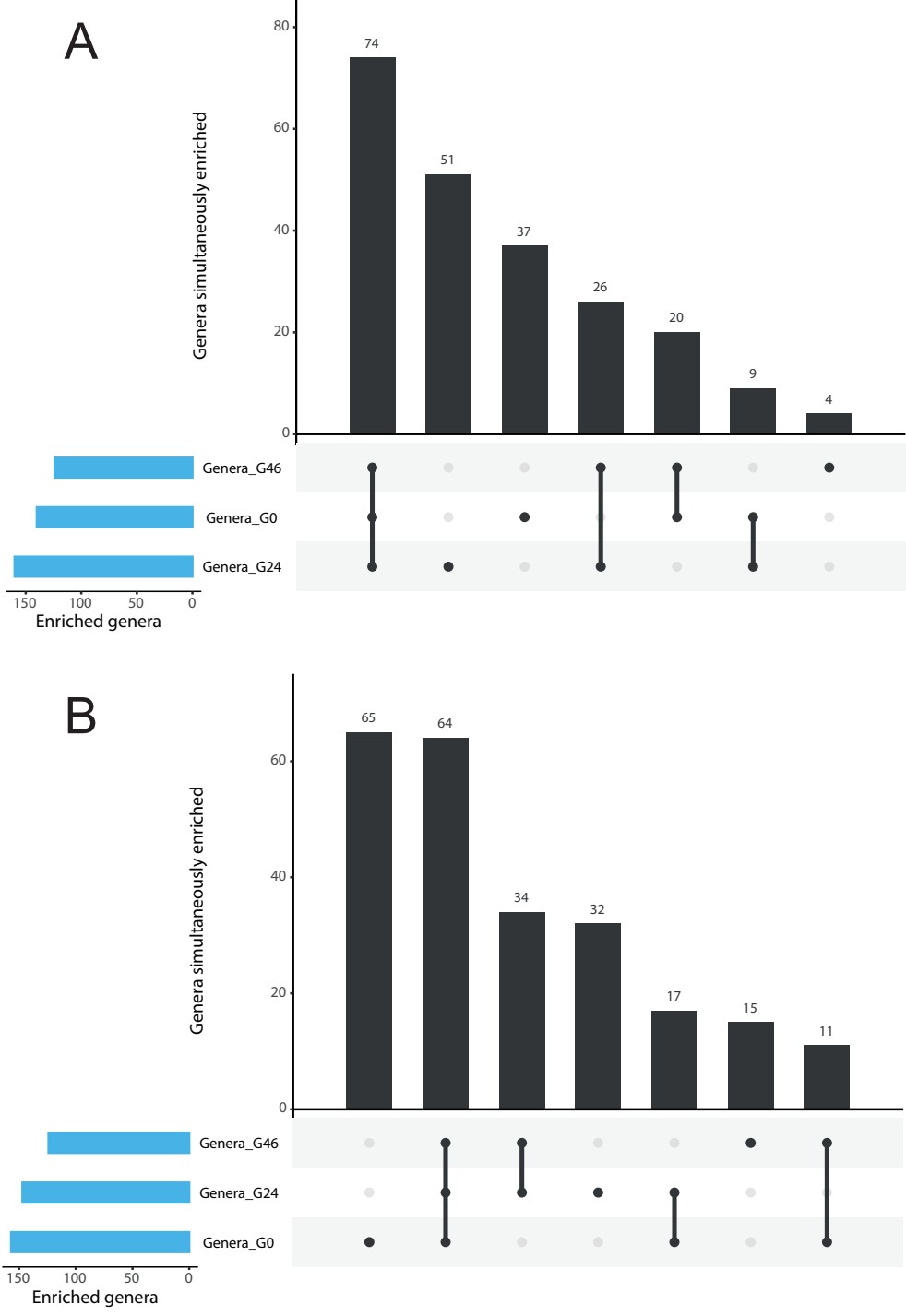

**Figure 3** **Gramine modulates bacterial abundances at the barley root-soil interface.** UpSetR plots of (A) Genera simultaneously enriched in pair-wise comparisons retrieved from unplanted soil and Morex rhizosphere, (B) and from unplanted soil and Barke rhizosphere, at the three levels of gramine tested. Vertical bars denote the number of genera enriched shared or unique for each comparison, while the horizontal bars the number of genera enriched in the indicated gramine concentration. In A and B genera differentially enriched at individual $p$ values $< 0.05$, Wald Test, FDR corrected.

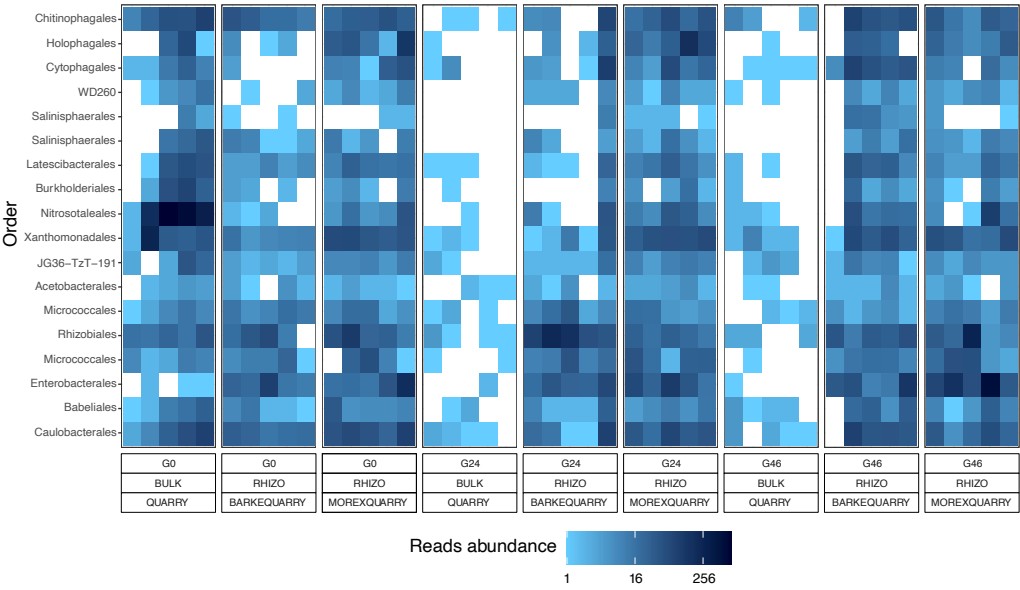

**Figure 4  Taxonomy and abundances of the gramine-responsive genera.** Heatmap of the 18 microbes, classified either at Order or Phylum level, significantly enriched in rhizosphere samples in a genotype- and gramine dosage- (24 and 46 μM, respectively) independent manner. Microbial enrichments defined at individual *p* values <0.05, Wald Test, FDR corrected.

rhizosphere- and gramine-responsive genera we observed a broad phylogenetic affiliation, encompassing members of the phyla Acidobacteria, Actinobacteria, Bacteroidetes and Proteobacteria. Next, we inspected how gramine application impacted on the genotype-driven diversification of the rhizosphere bacterial microbiota, using the number of genera (a) enriched from soil and (b) differentially enriched between the tested genotypes as a readout for this analysis. This analysis revealed that plants exposed to no gramine or the lowest dosage displayed a differential enrichment between cultivars of 31 and 42 genera, respectively (Figs. 5A and 5B, Wald test, individual *p* values <0.05, FDR corrected). Conversely, at the highest gramine dosage the host genotype effect was limited to 12 differentially enriched genera (Fig. 5C, Wald test, individual *p* values <0.05, FDR corrected). When we inspected the taxonomic affiliation of these differentially enriched genera, we made two observations. First, members of the phylum Proteobacteria accounted for the majority of differentially recruited genera (14 out of 31 genera at gramine 0 $\mu$mol L$^{-1}$, 24 out of 42 at gramine 24 $\mu$mol L$^{-1}$ and 6 out of 12 at gramine 46 $\mu$mol L$^{-1}$, respectively). The second observation was that despite the number of differentially regulated genera at the intermediate dosage of gramine was comparable to mock-treated specimens, the magnitude of the host selection was modulated by the host genotype itself, as Barke enriched for almost three times the number of Proteobacteria upon gramine application.

## DISCUSSION

In this study we demonstrated that the exogenous application of the indole-alkaloid gramine produced reproducible perturbations of the microbiota thriving at the barley root-soil

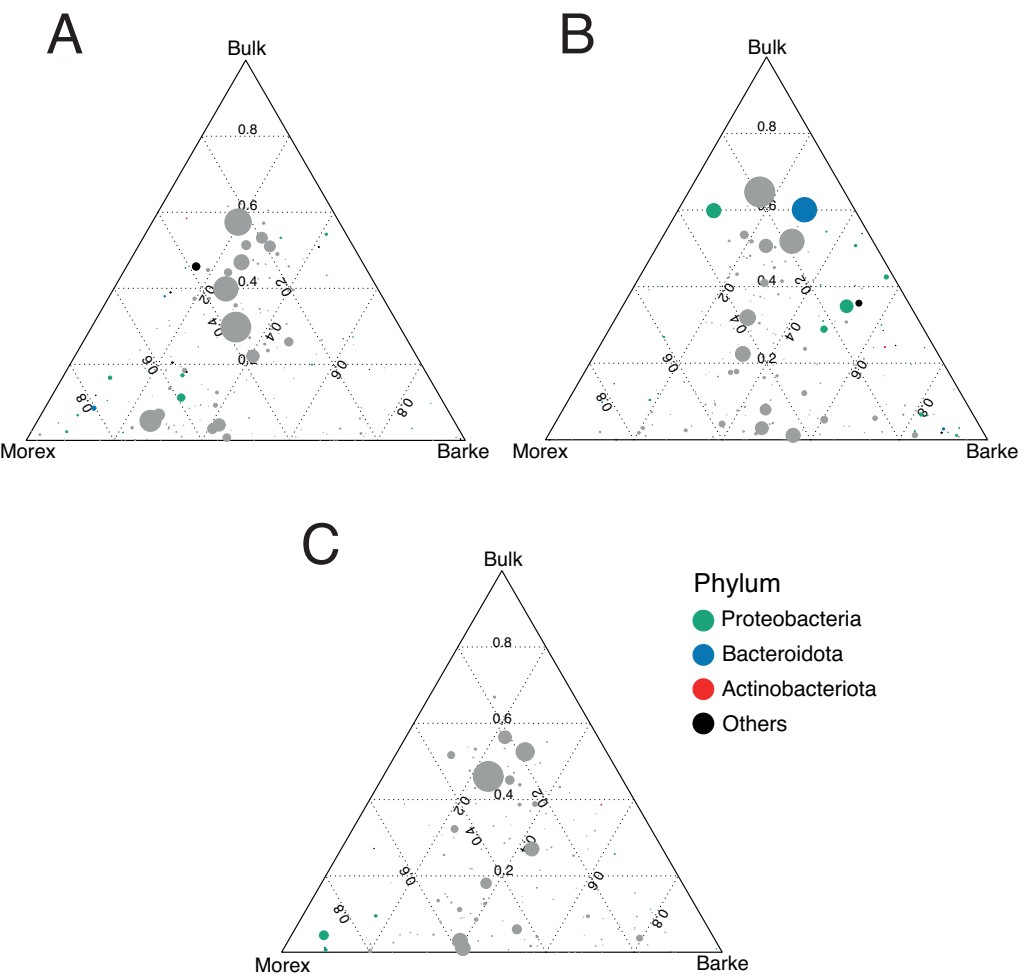

**Figure 5  Gramine application attenuates the genotype effect on the rhizosphere microbiota.** Ternary plots depicting bacteria distribution across the indicated microhabitats in sample exposed to (A) no gramine, (B) gramine 24 μM or (C) gramine 46 μM. In each plot, individual dots depict individual microbes whose size is proportional to their sequencing abundances. Position of the dots within the plots reflects the contribution of each microhabitat to microbial abundances. Coloured dots denote genera differentially enriched between Morex and Barke (individual *p* values <0.05, Wald Test, FDR corrected) color-coded according to their taxonomic affiliation at phylum level.

interface without triggering a discernable negative effect on plant growth performance on two barley elite varieties.

The observation that gramine applications impact on the soil microbiota is aligned with the findings of a recent survey performed by Schütz and co-workers monitoring the impact of the application of several plant secondary metabolites, including gramine, on bacterial communities of a German agricultural soil (*Schütz et al., 2021*). Similar to our findings, this study failed to identify an effect of gramine on community richness (*i.e.*, alpha diversity, Fig. 1), while observing a shift in community composition (*i.e.*, beta diversity, Fig. 2). Of note, the impact of gramine applications on individual bacterial enrichments displayed an experimental effect: in our investigation this was manifested with an apparent suppression

of bacterial proliferation in unplanted soil controls more pronounced than previously reported (*Schütz et al., 2021*). An alternative, and not mutually exclusive scenario, is that gramine acts as a substrate for the growth of other or additional bacterial members of the soil biota capable of outperforming taxa identified in this investigation.

In addition to differences in applications *per se*, it is important to consider that those experiments were conducted using different soil types. For instance, soil pH, one of the main drivers of bacterial community composition in soil (*Rousk et al., 2010*), in the two studies differed of ~0.5 unit. Although this property alone is unlikely to explain the differences between the studies, it may represent a contributing factor, alongside other parameters such as organic matter, to the differential impact of gramine applications on soil microbes. This concept is similar to what observed for the exogenous application to grassland soils of low-molecular weight carbon compounds, mimicking plants primary metabolites, altering microbiota composition in a soil- and substrate-dependent manner (*Eilers et al., 2010*).

Studies conducted with the model plant *Arabidopsis thaliana* contributed to define the impact of secondary metabolites on the composition and function of the plant microbiota. For example, the Brassicaceae-specific metabolites glucosinolates emerged as key regulators of the outcome of the symbiotic associations between *A. thaliana* and *Colletotrichum tofieldiae*, a fungal member of the Arabidopsis microbiota (*Hiruma et al., 2016*). Likewise, in an elegant association mapping study, Koprivova and colleagues identified a new nexus between a host genetic diversity and microbiota functions (*Koprivova et al., 2019*). In particular, genes underpinning the biosynthesis of the plant secondary metabolite camalexin emerged as regulators of sulfatase activities of the microbiota and its plant probiotic potential (*Koprivova et al., 2019*).

As *A. thaliana* is not a cultivated plant, we decided to compare the impact of gramine application on barley-associated communities with the one of other secondary metabolites identified in grasses such as maize, sorghum and oat which, similar to barley, have been exposed to the processes of domestication and breeding selection.

For instance, we identified a limited, but significant, effect of gramine application on the composition of the bacterial communities populating the root-soil interface (Fig. 3), indicating that gramine *per se* (or lack thereof) does not disrupt the capacity of individual barley genotypes of assembling a distinct rhizosphere microbiota. This is congruent with data gathered from studies conducted using maize lines impaired in the biosynthesis of benzoxazinoids grown in agricultural soils: despite mutants were capable of recruiting a distinct microbiota, this latter was compositionally different from the one associated with wild type lines (*Hu et al., 2018*; *Kudjordjie et al., 2019*). We consider these observations particularly relevant as benzoxazinoids are secondary metabolites produced by several grasses (*Frey et al., 2009*) with the important exception of the *Hordeum vulgare* clade (*Grün, Frey & Gierl, 2005*). Similar to gramine, benzoxazinoid display allelochemical, antimicrobial and insecticidal properties (*Niemeyer & Perez, 1994*; *Niemeyer, 2009*). A prediction of this observation is that, within the Poaceae family, different classes of secondary metabolites may have evolved to fine-tune microbiota composition. Congruently, sorghum produces a species-specific allelopathic compound

designated sorgoleone (*Czarnota et al., 2001*; *Dayan et al., 2010*) capable of selectively modulating bacterial microbiota composition as demonstrated by experiments conducted using RNA-interference lines impaired in sorgoleone biosynthesis grown under soil conditions (*Wang et al., 2021*). Likewise, oat plants impaired in the production of avenacin, a triterpenoid defensive compound active against fungal pathogens (*Papadopoulou et al., 1999*), recruit a taxonomically distinct rhizosphere microbiota compared to cognate wild type plants (*Turner et al., 2013*). Interestingly, the effect of avenacin manifested predominantly at the level of the eukaryotic component of the microbiota, particularly the protists Amoebozoa and Alveolata, rather than the prokaryotic counterpart (*Turner et al., 2013*). Although differences in the experimental and sequencing procedures existing among the aforementioned studies hinder the capacity of establishing first principles, these observations suggest that, in cereals, species-specific secondary metabolites act as a "gatekeepers" in the multi-step selection process proposed for the diversification of the plant microbiota from the surrounding soil communities (*Bulgarelli et al., 2013*; *Edwards et al., 2015*). The development of barley isogenic lines contrasting for gramine biosynthesis will be required to overcome a limitation of our investigation and ultimately prove (or disprove) these principles.

The impact of gramine application on the taxonomic composition of the barley rhizosphere microbiota revealed a relatively broad phylogenetic impact: members of 16 prokaryotic orders responded to gramine application in a dosage- and host genotype-independent manner (Fig. 4). Among those, of particular interest is the order Nitrosotaleales represented by the *Candidatus* genus *Nitrosotalea*. Member of this lineage have previously been characterized as ammonia-oxidizing archaea, *i.e.*, responsible for the rate-limiting step in the process of nitrification (*Treusch et al., 2005*), and, despite being autotrophic organisms, are capable of differential physiological responses in the presence of organic substrates (*Lehtovirta-Morley et al., 2014*). For instance, our data indicate that the application of gramine to unplanted soil communities suppresses the proliferation of this member of ammonia oxidizing archaea, pointing at a role of this compound in the biological inhibition of nitrification, as observed for others plant-derived compounds (*Tesfamariam et al., 2014*; *Kaur-Bhambra et al., 2021*). Yet, this scenario is difficult to reconcile with the observation that *Candidatus* genus *Nitrosotalea* is enriched in the rhizosphere of gramine-treated plants. A possible explanation could be derived by the niche adaptation of ammonia oxidizing archaea: barley seedlings, similar to other grasses, have a high rate of ammonia uptake and, in turn, this may create the optimum substrate conditions for the proliferation of organisms like *Candidatus* genus *Nitrosotalea* despite the presence of putative inhibitors (*Thion et al., 2016*). Congruently, previous experiments conducted with different barley genotypes identified ammonia oxidizing archaea among members of the resident rhizosphere microbiota in different soil types (*Glaser et al., 2010*). An alternative, but not mutually exclusive, scenario is that the gramine inhibitory effect is reduced by the activity of other microorganisms in the rhizosphere compared to the unplanted soil control. As the biological inhibition of nitrification has positive implications for sustainable crop production (*Coskun et al., 2017*), it will be interesting to further

investigate the relationships between gramine biosynthesis and ammonia oxidation in prokaryotes.

Conversely, when we inspected the impact of applications in a genotype-dependent manner we observed that the majority of microbes responding to gramine belong to the phylum Proteobacteria (Fig. 5). Members of this phylum also respond to differential exudation of benzoxazinoids (*Jacoby, Koprivova & Kopriva, 2021*), possibly representing another element of selection of cereal secondary metabolites towards the soil biota. As Proteobacteria represent the most abundant members of the plant microbiota across host species (*Hacquard et al., 2015*), the observed gramine effect may simply mirror the dominance of this group at the root-soil interface. Yet, a recent study conducted on maize demonstrated that Proteobacteria, specifically members of the family Oxalobacteraceae, promote lateral root density and shoot dry weight (proxies for plant growth) under nitrogen limiting conditions (*Yu et al., 2021*). As the soil tested in our experiments is limited in the availability of nitrogen for barley growth (*Terrazas et al., 2019*) and considering the enrichment of putative ammonia oxidizers archaea in the rhizosphere of treated plants (see above), it can be speculated that the differential Proteobacterial enrichment triggered by gramine application may be linked to nitrogen turnover in the rhizosphere. An additional observation derived from these experiments is that gramine application triggers a differential microbial recruitment in the two tested genotypes. A recent investigation conducted in tomato revealed that a component of the root exudates can be induced by the exposure to a given microbiota composition (*Korenblum et al., 2020*): it can therefore to be hypothesized that the differential compositional differences observed upon gramine application may be the result of a, genotype-dependent, fine-tuning of exudate profiles. An alternative, not mutually exclusive scenario, is a differential rate of gramine degradation in the rhizosphere (*Ghini, Burton & Gros, 1991*) of the two genotypes as this would alter the bioavailability of this compound to the resident members of the microbiota. Regardless of the scenario, it is interesting to note that gramine applications failed to trigger a sustained enrichment of members of the Actinobacteria, which can be considered as a hallmark of elite, gramine-free, barley genotypes grown in the same soil type (*Alegria Terrazas et al., 2020*) and in other modern/ancestral plant pairs (*Pérez-Jaramillo et al., 2018*).

## CONCLUSIONS

Our results indicate that the application of the indole-alkaloid gramine modulates the proliferation of a subset of soil microbes with relatively broad phylogenetic assignments. This effect is two-pronged: a component of the barley microbiota responds to gramine application in a genotype- and dosage-independent manner while other or additional host-derived mechanisms, underpinning the genotype diversification in the rhizosphere, modulate the effect of gramine application with a bias for members of the phylum Proteobacteria. As gramine biosynthesis has previously been reported as stress induced (*Velozo et al., 1999*; *Matsuo et al., 2001*), we anticipate that exposure to different soil characteristics, including different microbiomes, is likely to amplify (or obliterate) the effect of this metabolite on edaphic microbes. A limitation of our investigation was

represented by the lack of isogenic lines contrasting for gramine biosynthesis. We therefore propose to capitalise on these initial observations, and the expanding genomic resources for barley (*Maurer et al., 2015*; *Jayakodi et al., 2020*), to resolve the genetic basis of gramine biosynthesis and ultimately elucidate its adaptive value for plant-microbe interactions.

## ACKNOWLEDGEMENTS

We thank Malcolm Macaulay (The James Hutton Institute, UK) for the technical assistance in preparing the amplicon sequencing library and Rodrigo Alegria Terrazas (Mohammed VI Polytechnic University, Morocco) for the critical comments on the manuscript.

### Funding

The work presented in this article was supported by a Royal Society of Edinburgh/Scottish Government Personal Research Fellowship co-funded by Marie Curie Actions and a UK Research and Innovation grant (BB/S002871/1) awarded to Davide Bulgarelli. The funders had no role in study design, data collection and analysis, decision to publish, or preparation of the manuscript.

### Grant Disclosures

The following grant information was disclosed by the authors:
Royal Society of Edinburgh/Scottish Government Personal Research Fellowship.
Marie Curie Actions and a UK Research and Innovation grant: BB/S002871/1.

### Competing Interests

The authors declare there are no competing interests.

### Author Contributions

- Mauro Maver conceived and designed the experiments, performed the experiments, analyzed the data, prepared figures and/or tables, authored or reviewed drafts of the paper, and approved the final draft.
- Carmen Escudero-Martinez and James Abbott analyzed the data, authored or reviewed drafts of the paper, and approved the final draft.
- Jenny Morris performed the experiments, authored or reviewed drafts of the paper, finalised the library preparation and generated the sequencing reads, and approved the final draft.
- Pete E. Hedley performed the experiments, analyzed the data, authored or reviewed drafts of the paper, and approved the final draft.
- Tanja Mimmo conceived and designed the experiments, authored or reviewed drafts of the paper, and approved the final draft.
- Davide Bulgarelli conceived and designed the experiments, analyzed the data, prepared figures and/or tables, authored or reviewed drafts of the paper, and approved the final draft.

## DNA Deposition

The following information was supplied regarding the deposition of DNA sequences:

The sequences generated in the 16S rRNA gene sequencing survey are available in the European Nucleotide Archive (ENA): PRJEB39836.

## Data Availability

The version of the individual packages and scripts used to analyse the data and generate the figures of this study are available at GitHub: https://github.com/Stramon1um/gramine_microbiome.

## Supplemental Information

Supplemental information for this article can be found online at http://dx.doi.org/10.7717/peerj.12498#supplemental-information.

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
