# Peer review of "Applications of the indole-alkaloid gramine modulate the assembly of individual members of the barley rhizosphere microbiota"

_PeerJ, doi:10.7717/peerj.12498_

## Round 0.1 · original submission · Major Revisions

Thank you for submitting your work to Peer J and specifically to the special topic on Animal and Plant_Microbe interactions biology.
Although with some delay (mainly due to the holiday period we had, which always causes delays in the reviewing process), we have now received the reviewers' comments. A number of issues have been raised that need your attention; especially some methodology aspects that are not clear as currently described as well as claims and discussion points that need to be backed up (or re-stated) to add clarity to your manuscript. I concur with the reviewers on these and suggest taking them all into consideration in a revised manuscript.

·

Basic reporting

The English could be simplified in a few points (see additional comments section) in order to improve the text flow... but overall the English is of high quality.

Also there are discussion points in the results (see additional comments section) which need to be rephrased/removed.

Otherwise, the manuscript meets the journal standards.

Experimental design

The authors have not included any gramine producing wild-type and its gramine deficient mutant in their study… an addition as such would have greatly enhanced their discussion. I would suggest to discuss this event in the least.

Validity of the findings

There are some results that are somewhat overstated, but this could be addressed through text changes. See the "additional comments" section for more details.

Additional comments

Herewith, I provide my views on the manuscript entitled “Applications of the indole-alkaloid gramine shape the prokaryotic microbiota thriving at the barley root-soil interface” by Maver et al describing the effects of the allelopathetic gramine on the prokaryotic microbial community. The authors have used a bulk soil vs two gramine deprived varieties, and endeavoured to understand the effects of gramine by contrasting three different gramine application levels (0, 24 and 48 µmol L-1) of applied gramine.

General comments:

I believe that there is merit in this study particularly for the portion of the research community interested in the applied aspects of alkaloids. I also think that the authors did a meticulous work in several parts of the experimental design although they could have improved their setup as I will explain below. Finally, The English is of high level, although sometimes difficult to follow due to the use of very literate language (I like it in most cases).

Having mentioned the strong points of this work, I would like to mention some weaknesses (which are less important but could be easily addressed by toning down the discussion or acknowledging the associated pitfalls).

E.g. one weakness is the fact that the authors have not included any gramine producing wild-type and its gramine deficient mutant in their study… an addition as such would have greatly their enhanced discussion. I would suggest to discuss this event in the least.

Another weakness is that the authors seem to consider gramine solely (or in the most manuscript) as a toxicant against prokaryotes (although gramine has been studied mostly for its untifungal properties). They do not consider the fact that gramine could also be C/N source for prokaryotes, leading to the manipulation of the microbial composition due to changes in the nutrient balance… or gramine could be toxic against fungi and, hence, affect their degradation. There are some comments in the end of the report about indirect effects but most of the document passes the feeling to the reader that graine is toxic against all tested prokaryotic members.

I see highly meticulous attitude in some aspects of the experimental design (e.g. negative controls for all indexes used in multiplex sequencing, several controls in other design aspects etc…) and quite relaxed attitude in some other parameters (e.g. 2x150 in >300bp amplicons)

Finally (less important point… but needs to be addressed), there are some discussion points made in the results section… I would suggest changing or removing these.

Specific comments:

Introduction

L51: “support” instead of “supports”

L58: I am not sure the authors want to use the word “checkpoints”… maybe “milestones”… either way the sentence needs rephrasing.

L66-87: Here the authors apparently want to state that domestication is related to the production of allelopathetic metabolites (like gramine) which may contribute to the manipulation of the rhizosphere microbial communities. I would suggest adding such an opening sentence in the beginning of this part that will immediately make clear to the reader what the two paragraphs below are about. Besides, these paragraphs are the most important for introducing the aims.

Material and methods

L130: media? Only agar? Else, what was the composition? A reference could support as well.

L138-139: what was the basis of the choice of the gramine concentration? Is it close to the naturally occurring concentration?

L207-210: This strategy barely leaves any chance for amplicon reconstruction (the amplicon plus the barcode is more than 300bp long while the 2x150 cover just 300 allowing no reliable overlap for amplicons to be reconstructed). Hence, no second strand verification is provided (at least at the terminal, most error prone, parts of the sequence reads) by this strategy and the quality filtering relies solely on estimated errors by the base-calling software rather than physical evidence. This approach has been published but does affect the confidence by which these data should be considered.

L219: please provide a link or a reference for the pipeline

L222: high quality? Please provide a brief description of the criteria… I remind that since no overlap exists there is no second strand verification of the read-end error prone sequence parts. I do accept the fact that this approach was published in the past, but I would like the authors to acknowledge its weaknesses.

L227: I might be missing something, but, read merging is not possible for the vast majority of the amplicons obtained with this primer set (~300bp amplicons with some length variance) and given the sequencing-approach (2x150 = 300bp per amplicon, hence no overlapping bases between the reads)… the authors mast mean that they used the parameter allowing concatenation of the reads in case no overlap is obtained… is that right? Please describe it clearly.

L232: The authors mean “screened” or “ASVs of non-target taxa were filtered out” instead of pruned. The authors seem to be literate but at some points their expressions are somewhat odd.

L237: I agree with this approach (i.e. probably the authors wanted to retain sequences that were clearly associated with the treatments)… however, it has to be justified. Please provide the reasoning.

Results

L277: the title looks like a conclusion… since this belongs in the results section, I would suggest that the subsection titles be more descriptive

L314-317: This sentence looks to be more of a discussion statement than a results description.

L322: look at my comment for the previous subsection title.

L331: “respond” seems to be a strong statement… I would suggest using “coincide” or similar

L334: same as above for “responsive”


L356-361: This belongs in the discussion not the results. I would also suggest toning down statements like “otherwise suppressed by gramine application, appeared rescued”. The authors did not provide or cite any mechanistic information about possible toxicity of gramine on microbial cells, let alone particular taxa. E.g. gramine itself may act as a nutrient and therefore selection means as do root excretes of nutritional value. Without a gramine producing wild type control (which I understand is difficult to find in a way that it will have a similar genotype with the tested plant varieties), these statements seem a bit far-fetched.

Discussion

L378: Or may act as nutrient that causes outperformance of responsive microorganisms over others that lower their relative abundance which the authors may assume to be suppressed by gramine.

L381: a 0.5 pH difference seems marginal given between lab differences in experimental performances… I would rather avoid discussing it as an important difference between experiments. What about the organic matter (which is also a major microbial diversity driver – actually it is the most important one)?
L393: from a study… or … studies

L394: there the authors used benzoxazinoid impaired mutants along with the wild type… something that could help enhancing the impact of this study as well (i.e. the use of wild-type and gramine impaired wild-types next to the tested varieties).

Reviewer 2 ·

Basic reporting

Maver and co-authors analyzed the changes in barley rhizosphere and bulk soil bacterial communities after addition of the secondary metabolite gramine to soil in a pot experiment. Gramine is present in root exudates of wild genotypes, but absent in some modern varieties, such as Morex and Bark analyzed in the present study. This topic is important, since understanding the exudate molecules that shape the rhizosphere microbiome is essential to guide plant breeding and management strategies aiming to explore the beneficial plant-microbe interactions for more sustainable agricultural systems. The manuscript is relatively well written and organized, but requires grammar corrections in many sentences. The literature is in general well referenced, needing just a few inclusions (e.g. acknowledge other factors that affect the rhizosphere microbiome selection in addition to exudates - Intro; mention other secondary metabolites that shape the rhizobiome such as coumarins, glucosinolates and camalexin - Discussion). The figures are in high quality and the legends are clear.

Experimental design

The hypothesis of the study is clear and methods are standard in the field. The gramine adsorption assay was particularly interesting to make sure this compound is available in the soil solution to influence the soil bacteria. However, the experimental design would be more appropriate to test their hypothesis if the modern barley genotypes absent in gramine exudation were compared to the wild genotypes able to exude gramine. More genotypes would be needed for a stronger statistical power, avoiding bias associated with specific genotype background. The methods are properly described for replicability, but sometimes too much detail is provided, please summarize (e.g. sequence analysis). Moreover, the Methods should not be organized with checkpoints, as it is in some parts (Lines 185-189, 215-252). Removing the ASVs with <20 reads in at least 11% of samples (Lines 237-239) is too stringent, a lot of taxa was possibly lost in this process. Where did authors find this approach? Please add reference.

Validity of the findings

Although the authors claimed in the title and throughout the manuscript that the microbiota was affected by gramine addition, the results showed the opposite. Permanova results indicated that gramine concentration did not affect the bacterial community structure (Table 1) and the CAP showed a clear separation of samples according to gramine concentration only for the bulk soil (Figure 2). Observing specific differences in the relative abundance of a few bacterial genera does not mean the prokaryotic microbiota was shaped by gramine concentration. The greater effect in the bulk soil than in the rhizosphere beta diversity found here does not support the idea that gramine is a selecting factor for the barley rhizosphere microbiome, despite a previous study showed a significant effect (Schütz et al., 2021). The authors should change the focus of the manuscript to better reflect their findings.

Reviewer 3 ·

Basic reporting

Maver et al. tested the effect of exogenous application of gramine (two doses) on bacterial communities of two modern barley cultivars and compared to unplanted soil (bulk soil). Their conclusion is that gramine can impact the bacterial communities in the rhizosphere of barley, with some enriched taxa.

The submitted paper conforms to professional standards as regarding to the use of English. The references are sufficient and provide a good overview of the topic. The provided figures are relevant to the understanding of the paper. Raw data is available at ENA and I liked the way they made the scripts available at Github, so their results can be reproduced.

Experimental design

The research is original and methods are also in accordance with what people are currently using (DADA2, ASVs).
Methods are briefly described and the full information can be seen in Terrazas et al. (2019). However, it's a biorxiv paper. I would recommend to add the full details in the present paper if there is more information to be added.
I could not see the script using phyloseq, vegan and deseq2 in R, regarding the removal of mitochondria, chloroplasts, filtering low abundance of ASVs, agglomeration of ASVs at genus level, alpha diversity analyses, PCoA, CAP, Permanova and deseq2 analysis. Please add it to your Github repository.

Validity of the findings

I have one question, related to the choice of cultivars. Why did you not compare the application of gramine in both modern and wild barley cultivars? It would have been interesting to see the differential bacterial recruitment between elite x wild. Perhaps include somewhere in the text why this comparison was not made.

Additional comments

Minor comments:

Line 64 – remove the comma before metabolites

Lines 124-125 - Missing reference – International Barley Genome Sequencing Consortium, 2012)

Line 240 – how were ASVs agglomerated at Genus level? Was it done manually? Please provide additional information regarding the way it was done.

Line 332 – In the text you say 18 genera, but looking at figure 4 it’s taxa mostly at the “order” level. Correct that in the text. Is there any phyla there?

Line 420 – “17 prokaryotic orders”. Weren’t there 18?

Table 1 – I would suggest you don’t need that table as you provided the same information in the text (lines 310, 312, 314).

Figure 3 – I liked the idea of the upsetR plot, however, it requires further explanation. When you show the number of enriched genera (comparison between bulk soil and Morex – A) and (bulk soil and Barke – B), do the numbers belong to enriched genera in the rhizosphere or in the bulk soil? E. g. By looking at the plot you can say that when you apply G46 you have 4 enriched genera. But in which niche? Are they enriched in both bulk soil and rhizosphere of Morex? If that is the case, would it be possible to show which ones are enriched in bulk soil and rhizosphere, separately? In that way we could see the rhizosphere effect.

Figure 5 – In the legend, what is the taxa correspondent to gray?

---

## Round 0.2 · accepted · Accept

There are no further comments to be addressed to this manuscript.

·

Basic reporting

The authors have addressed all my previous comments clearly. All field associated parameters comply with the journal standards.

Experimental design

All field associated parameters comply with the journal standards.

Validity of the findings

All field associated parameters comply with the journal standards.